# The geographical dynamics of global R&D collaboration networks in robotics: Evidence from co-patenting activities across urban areas worldwide

**Thomas Scherngell**[1]*, **Katharina Schwegmann**[1], **Georg Zahradnik**[2]

**1** AIT Austrian Institute of Technology, WU Vienna University of Economics, Vienna, Austria, **2** AIT Austrian Institute of Technology GmbH, Vienna, Austria

* Thomas.Scherngell@ait.ac.at

**Data Availability Statement:** Data cannot be shared publicly because the RISIS research infrastructure does not permit for commercial use.

## Abstract

The focus of this study is on the geography of robotics Research and Development (R&D) activities. The objectives are, *first*, to identify hotspots in robotics R&D worldwide, and *second*, to characterise structures and dynamics of global robotics R&D collaboration networks through detailed geographical lenses of global urban areas. We use patents as marker for R&D activities, and accordingly focus on technologically oriented R&D, drawing on information from patents applied for between 2002 and 2016. We employ an appropriate search strategy to identify relevant robotics patents based on detailed levels of the Cooperative Patent Classification (CPC) and assign patents to more than 900 global urban areas based on the inventor addresses. The co-patent networks are examined from a Social Network Analysis (SNA) perspective by means of robotics co-patents, contributing to a global network where urban areas are the nodes inter-linked by joint inventive activities recorded in robotics patents. Global SNA measures illustrate structures and dynamics of the network as a whole, while local measures indicate the specific positioning and roles of urban areas in the network. The results are original in characterising the global spatial emergence of this generic new industry, highlighting prominent urban hotspots in terms of specialisation in robotics R&D, pointing to a global shift reflected by the increasing role of emerging economies, in particular China. The global robotics R&D has grown significantly both in total patenting and also in terms of R&D collaboration activities between urban areas. Also, for the networks, growth is not equally distributed, but is rather characterised by significant spatial shifts, both in terms of cities declining or climbing up the specialisation ranking, but even more in terms of the spatial network structure.

## 1 Introduction

New knowledge created by Research and Development (R&D) activities is widely considered as the basis for successful innovations [1]. Nowadays, the creation of new knowledge is

Requests for RISIS Patent data access may be directed to RISIS (https://rcf.risis2.eu/datasets).

**Funding:** TS, KS, GZ: partly funded by the European Union under Horizon2020 Research and Innovation Programme Grant Agreement n° 824091; https://ec.europa.eu/info/funding-tenders/opportunities/portal/screen/programmes/h2020; funder played no role in study design.

**Competing interests:** The authors have declared that no competing interests exist.

increasingly accomplished within a complex web of interactions between researching organisations [2], often referred to as R&D collaboration networks (see, e.g. [3, 4]). Looking at knowledge creation processes and underlying network structures from an economic geographical perspective, the *geography of innovation* literature is the most important one as it stresses the spatially localised nature of knowledge creation and is hypothesised to be one of the main explaining factors for divergent spatial economic development [5]. Against this background, we can observe an enormous growth of empirical studies over the last two decades that investigate the geography of knowledge creation, R&D activities, networks and innovation, related to new upcoming datasets enabling empirical research to capture different types of knowledge and R&D outputs (see [6] for a recent overview).

However, most of these works tend to look at knowledge creation and R&D collaboration networks from an aggregated perspective, in particular in terms of different underlying knowledge domains or technological fields [7]. Therefore, interest in tracing the geography of knowledge creation and R&D collaboration networks for specific relevant technological fields has grown tremendously, also, for instance, in context of the revitalised debate on mission-oriented research policies [8]. In the latter context, the focus does not lie on classical economic sectors, but on cross-cutting, generic technologies that are not only of crucial importance for future economic competitiveness, but also of specific relevance for addressing important societal challenges, such as climate change, health, or ageing societies.

In this study, we focus on *robotics* as one central technological field addressed by mission-oriented research policies–for instance in a European policy context which is part of the Key Enabling Technologies (KETs) concept, and the related definition of General Purpose Technologies (GPT) (see, e.g., [9]). We shift attention to geographical dynamics of R&D activities and related collaboration networks in the field of *robotics*. Clearly, robotics is not only considered as an important breakthrough technology that is substantially contributing to the growth of the world economy related to remarkably increasing demand [10], but for some major aspects has the potential to directly responds to some of the societal challenges such as demographic ageing or environmental threats. On top of that, robotics is even considered to comprise the ability for structurally transforming whole economies and societies, though not only in a positive way in terms of added economic value and new societal opportunities, but also with potential problematic consequences, for instance, increased unemployment due to drastic transforming labour markets. In any way, robotics will become pivotal for the economic competitiveness of regions and countries, since improved capabilities of robots and the benefits derived from the implementation in the production cycle accelerate the demand across many industries, causing significant growth potentials and productivity gains.

When considering the rising importance of robotics and respective R&D activities, the general trend towards more intensive collaborations in R&D has to be taken into account in any empirical investigation. This is of particular importance for this technological segment since the robotic industry is highly research and capital intensive requiring a lot of expertise and knowledge that increasingly cannot be covered by single organisations, or even regions and countries [11]. However, although the robotics segment is expanding constantly and becoming more relevant in economic and societal terms, only few studies exist that systematically trace the global landscape of robotic R&D activities from a geographical perspective, both in terms of R&D hotspots and the related global R&D collaboration networks behind it [12].

Being considered as one of the most important breakthrough technologies of our times, providing fundamental insights into the changing geography of robotics R&D is a gap in the geography of innovation literature. In addressing this gap, the focus of this study is on characterising the global spatial emergence of this generic new industry. The overall objective of this study is to identify and empirically characterise the changing global R&D landscape in robotics

from a geographical perspective and at a very detailed geographical level while combining a spatial technological specialisation perspective with concepts from R&D collaboration networks. *First*, we want to identify global R&D hotspots specialised in robotics R&D at a detailed geographical level of urban areas while having in mind that R&D is quite unevenly distributed within countries. *Second*, we intend to trace global robotics R&D collaboration networks for the first time, illustrating the dynamics within the network on a global scale from the perspective of urban areas. Moreover, we aim to enrich and relate to theoretical debates on the geographical dynamics of R&D activities and related collaboration networks, both from the perspective of the literature on the geography of innovation and R&D internationalisation, but also in a network science context. Empirically, we use patents as marker for R&D activities and accordingly focus on technologically oriented R&D with the clear goal of economic commercialisations, rather than basic, fundamentally scientific oriented R&D. Specifically, patent-based indicators are extremely useful for comparing and monitoring trends in the technological output, as well as in the context of tracing R&D collaboration by means of co-inventions, i.e. patents featuring inventors located in different urban areas or countries [13].

In its empirical strategy, this study draws on information recorded in patents applied for between 2002 and 2016. We employ a search strategy to identify relevant robotics patents based on detailed levels of the Cooperative Patent Classification (CPC). We assign the patents to more than 900 global urban areas based on the inventor addresses given in the patents and their geocoding. To identify R&D hotspots and global specialisation patterns, we calculate a Revealed Technological Advantage (RTA) index measuring the level of the urban area's relative technological specialisation in the robotic segment. The R&D collaboration networks are examined from a Social Network Analysis (SNA) perspective, following previous research [14]. Here, we trace robotics co-patents, also observed for the time period 2002 to 2016, giving rise to a global network where urban areas are the nodes inter-linked by joint inventive activities recorded in robotics patents. Global SNA measures illustrate structures and dynamics of the network as a whole (e.g., size of the network, intensity of collaborations), while local measures indicate the specific positioning and changing roles of urban areas in the network.

The remainder of the study is organised as follows. *Section 2* clarifies the relevant theoretical and conceptual foundations of R&D and related collaboration networks in the context of robotics as important breakthrough innovations. *Section 3* is devoted to the methodological approach for analysing the global R&D landscape in robotics, introducing in some more detail the empirical setting, including the patent data used, the study area, the time frame as well as the analytical approaches in form of the RTA and the SNA analysis. The empirical results of the study are discussed and presented in *Section 4*, starting with an illustration of the findings of the RTA analysis in terms of global R&D robotics hotspots. *Section 5* then discusses global robotics network structures and dynamics based on the SNA including illustrative network visualisations at the level of global urban areas. *Section 6* concludes the study by summarising and reflecting on the most essential results and findings and finalises with possible research gaps and future research ideas.

## 2 Conceptual background and literature review

In what follows we discuss in some more detail the relevant theoretical and conceptual foundations of this study. Initially, we describe the main characteristics of robotics as a breakthrough innovation, both in an economic but also a social context, as well as the specific characteristics of innovation processes in robotics. Then, R&D collaboration networks are specified as important channels for knowledge flows, assumed to be specifically relevant for highly knowledge intensive fields like robotics, discussing specific geographical and network structural mechanisms shaping such networks.

## Robotics as a breakthrough innovation

In recent years, both the global economic and societal demands for robotics have been increasing remarkably [15]. The market for industrial robots has been growing significantly at rates in double-digit numbers per annum after millennium (see, e.g. [10]). We define *robotics* as the segment of automation *technologies* replacing human efforts [16]. This involves all kind of physical machines or devices programmed to perform a variety of different tasks, with some level of interaction with the environment, but with limited or no input from a human operator. According to the International Federation of Robotics (IFR) and in line with the international standard ISO 8373, robots can be, based on their intended application, classified into industrial and service robots (see [17, 18] for further definitions of different classifications for robots). In 2019, industrial robots operating in factories around the world reached a record number of 2.7 million (mainly applied in handling and assembling in automotive, food and plastic industries) which is an increase of 12% compared to 2018 [17]. Main drivers of this growing demand are robots' improved capabilities, their widespread potential use, productive activity and falling costs [19]. In addition, manufacturing processes are getting more complex as the number of customised and diversified products rises. This makes robots highly attractive for increasing efficiency and flexibility within the production process [15]. In the latter context, the fast development of artificial intelligence technologies–e.g. used for patterns recognition or automatised communication–integrated in robotics applications strongly contribute to increasing decision-making capabilities [20, 21], and accordingly the attractiveness of robots, mainly in the manufacturing sector but also in service industries [22].

However, in terms of social consequences, the increasing role of robotics and the expectations resolving from it are much more mixed as what concerns positive versus problematic or negative societal impacts. For instance, robotics may–particularly in a short-term perspective–have negative social effects, in particular in labour markets and their transformation. Labour shares may decline relatively in industries that are more amenable to automation, leading to increasing unemployment rates [23, 24]. However, at the same time new jobs will be created which will require more complex and intellectual skills that cannot be acquired by former industrial workers, and moreover may occur in other geographical locations as well [25]. This transformation process has to be observed critically and accompanied by respective dedicated and policy-relevant future research. For the study at hand, the enormous recent economic growth of robotics is the main background being of relevance for the subject under consideration.

In fact, investments in robotics are expected to increase even further across all industries, especially in the automotive, electronics and pharmaceutical sectors, but also with an enormous potential to disrupt existing economic and social facets of everyday life, considering robotics technologies as one of the most important breakthrough innovations nowadays [12]. In this context, two inter-related factors are specifically important illuminating the robotics sector: *First*, robotics is a highly research-intensive, dynamic, and complex technological segment [11], requiring particular expertise, knowledge endowments, skills and specialisations as the products in the robotic market are ranging from classical industrial robots to fully autonomous service robots. In reference to the findings of [26], this directs per se to an increasing demand for collaborative arrangements among specialised robotic companies, given the diversified pieces of knowledge (e.g., motive power, control system, sensing) to be integrated in the R&D and innovation process [11]. *Second*, given the high R&D intensity, on the one hand, and the combination of the growing demand and economic potentials, on the other hand, robotics technologies are to a substantial degree subject to public R&D funding initiatives across a number of countries worldwide, but also for supra-national organisations such as the

European Union (EU). [17] shows that most countries invest in robotics: China for example is seeking to achieve a major breakthrough for key components and tries to enhance robots' integration with new-generation information technologies while having a budget of 577 million dollars (2019). Japan planned to invest 351 million dollars (2019) in R&D projects and focuses for instance on the development of medical and supporting devices for the aging population. Finally, the EU's key target is to digitalise the industry with the application of robotics technologies.

## Public R&D funding and R&D collaborations

Such public programmes come into play as they are considered as substantial drivers for innovations, for competition in the industry and for economic productivity growth in general (see, e.g. [26, 27]). Moreover, public R&D programmes have recently shifted their focus from single actors to joint consortia in dedicated R&D projects, with the aim to sustain knowledge circulation in the regional or national system of innovations. These R&D collaborations can be viewed as a dynamic network whose structure develops and deepens over time. Related to this, increasing technological complexity and challenges derived from globalisation, such as higher competition and a fast-changing environment, accelerate the need for international cross-border R&D collaborations [28, 29]. Three main arguments are stressed in this context: *First*, expertise and knowledge are shared and transferred into the R&D project by the collaboration partners. Taking Schumpeter's perspective, the combination of external knowledge creates new knowledge among researchers that increases the quality and output of R&D in the collaboration network [30]. *Second*, the increasing complexity and challenges of innovation processes make it almost impossible to be specialised in every segment (see, e.g., [31])–an issue of particular relevance in the case of robotics based on the characterisation of the field put forward above. Therefore, collaborations among R&D actors are needed to pool and share resources, such as knowledge, machines, and equipment. *Third*, researchers can take advantage of the network that is created or already exists among the organisations. They learn and accumulate new knowledge from constantly shared and reviewed knowledge and information in the network [32].

In this context, it is important to stress some of the arguments from theoretical and empirical literature on drivers for R&D collaboration networks, in particular from a geographical perspective which is also the focus of this study. [6] provides a compact overview on the empirical literature that investigates the geography of R&D collaboration networks of different kind, including networks reflected in co-patents. It is argued that different types of proximity [33] between two regions are conducive for collaboration between them, or vice versa hamper collaboration when two regions are less proximate to each other. Eventually, this will lead to differing collaboration intensities which are often estimated by means of spatial interaction modelling approaches. Overall, empirical studies indicate that the majority of R&D collaborations are still geographically localised, with co-patent networks being even more localised than, for instance, scientific or project-based networks. Further, it is emphasised that technological effects and cultural and institutional barriers are–next to geographical distance–important dimensions for shaping the characteristics and structural dynamics of these networks at the regional level, in particular differing cross-region collaboration intensities. More recent works (see, e.g. [34, 35]) stress other important concepts being of relevance for determining R&D collaboration networks, such as the availability of complex knowledge or related capabilities in two regions or countries.

Besides these considerations which are mainly driven by the geography of innovation literature, network science provides important spatial arguments on the shaping of R&D

collaboration networks from a pure network analytical perspective [4]. Concepts like preferential attachment or hub-and-spoke structures are key in this context, assuming that in networks certain structures are likely to occur due to purely network structural mechanism. For instance, spatial areas may more likely increase collaborations to other areas showing similar network attributes, e.g., in terms of their number and quality of collaboration links. In Social Network Analysis (SNA), this is usually referred to as homophily, i.e., social actors are more likely to interlink when they have similar attributes [36]. Moreover, the positioning of individual spatial areas in a network in relation to other areas is an important driving mechanism of the network as a whole, and also an important analytical dimension of the study at hand by means of the network centrality concept employed in the empirical analysis (see Section 4).

The empirical analysis of the study at hand lies in the tradition of the literature stream discussing the evolution of R&D collaboration networks jointly in terms of their spatial configurations and related underlying network structural mechanisms (see, e.g. [4, 37, 38]). Though we take a descriptive perspective, for instance, by looking at spatial shifts of R&D hotspots and related changing configurations in the underlying network structures (see Section 4 and Section 5), we add to these debates underlining for a specifically relevant technological field and show that clearly different network structural mechanisms may be at stake shaping the territorial dynamics of innovative activities. We hypothesise that the global R&D network in robotics has grown over the past decades, but that it is also highly dynamic in terms of new actors, and accordingly new geographical locations come into play. In what follows we illustrate our empirical setting and the detail out the methodological approach, before we present an empirical characterisation of the global robotics R&D landscape from the perspective of hotspots and R&D collaboration networks.

## 3 Data and methods

The empirical strategy employed in this study follows a large body of previous empirical research using international patent applications for empirically tracing R&D activities and collaboration networks (see, e.g., [27, 39, 40], among many others). Patents grant a property right securing the patent owner a return on R&D investments [20], and are widely used indicators in innovation research as marker for new knowledge resolving from R&D activities with a commercial application perspective ([41, 42]). In our empirical approach, we mobilise the location of inventor addresses given on patent documents for tracing R&D hotspots, on the one hand, and inter-regional co-inventive activities, referred to as co-patents, for tracing R&D collaboration networks, on the other hand. Co-patents are defined as patents featuring at least two different inventors in two different regions (here urban areas), giving rise to an inter-regional R&D collaboration activity for the patent under consideration. Note that using co-patenting ensures the tracing of a very specific form of commercially oriented, technological collaborative R&D, while other indicators such as co-publications or joint R&D projects are more indicative of scientific collaborations. In the robotics context, patents seem to be specifically suitable as robust indicators for new relevant knowledge because the field is highly research-intensive which requires intensive capital expenditures and R&D often takes place several years before commercialisation [12, 43].

The patent data in this study is traced from the RISIS Patent database (spring 2020 edition), available from the RISIS research infrastructure. RISIS Patent derived from the EPO PAT-STAT database is specifically relevant for tracing geographical dynamics of knowledge creation. In RISIS Patent, an address is identified for 75% of inventors (to be compared with 10% in the initial raw EPO PATSTAT data) and 67.4% of the addresses are geocoded and associated to 'functional areas' (urban and rural) worldwide, mobilizing the RISIS CORTEXT geocoding

service [44]. One of its most interesting features–in context of our research focus–is the very detailed geocoding of inventor addresses to functional urban and rural areas worldwide put forward by a group of geographers as introduced by [45], or in a similar context in [46–48]. The urban areas used as spatial configuration in this study have been built based on several sources of information, including 667 functional urban areas in Europe [49], 17 urban areas for European associated countries [50], 104 functional urban areas for non-European OECD countries [46, 51], 450 urban areas in the US [52], 375 Chinese urban areas [53] and 2,587 urban areas for the rest of the world [54, 55]. These approximately 4,200 urban areas have been combined in one spatial layer for a global perspective. Then, a regional layer put forward by [55] has been used to fill portions of geographical space which are outside urban areas (referred to by the prefix "Other" in the labelling of spatial units). Note that such spatial units can be very large when the population is rather distributed in a number of cities below the threshold of 50,000 inhabitants.

The unit of observation of the RISIS patent database are priority patents, the very first patent applications worldwide to protect the respective invention. Moreover, it has standardised information on patent applicants by employing semi-automated cleaning of applicant names, mainly companies (see [44] for details on the construction of the database). Based on [12] we employ a patent mapping strategy that uses 13 CPC classes to extract the patent data from the RISIS patent database (see Table A1 in File for a list of the CPC and a respective description of the technological content). CPC (Cooperative Patent Classification) is an extension of the International Patent Classification (IPC) system and an internationally organised patent classification system suitable for searching patent applications in robotics. Its unified system is organised in a hierarchical way (the technology fields are separated into nine sections with approximately 250.000 subdivisions). Data has been extracted for the priority years from 2002 on and was grouped into three different time periods: (1) 2002–2006, (2) 2007–2011, and (3) 2012–2016 for the longitudinal analysis. In total, the sample dataset includes 47,281 inventor locations listed in 18,184 patents, with 41,397 of them assigned to 933 urban areas that locate at least one inventor in one of three periods. Geocoding tools have then been mobilised to assign all patents to more described spatial configurations of urban areas, based on the inventor addresses given in the patents (using the Cortext tool, see www.cortext.net).

In our methodological approach, we use these patent data to identify global R&D hotspots and specialisation patterns in robotics. Regional specialisation in robotics exists when a spatial unit shows a comparatively high proportion of inventive capacities in robotics, measured by the Revealed Technological Advantage (RTA) index [56, 57].

The RTA is defined as

$$RTA_{ik} = \frac{p_{ik} / \sum_{k=1}^{m} p_{ik}}{\sum_{i=1}^{n} p_{ik} / \sum_{i=1}^{n} \sum_{k=1}^{m} p_{ik}} \tag{1}$$

where $p$ denotes the number of patent applications, $i$ the urban area ($i = 1, \ldots, n$) and $k$ the technology class robotics (being the sum of the CPC codes listed in Table A1 in File). Accordingly, the RTA index is the ratio of the regional share of patenting in robotics divided by its regional share of total patenting in all sectors [58]. Note that the number of patents per area is based on fractional counting of inventors, i.e., the patent is divided equally among all areas in which inventors of the patent are located (e.g., if a patent with two inventors located in two regions A and B, it counts a fraction of 0.5 for both regions). The index is equal to one when the share of robotics is exactly equal to the share of robotics patenting worldwide; smaller than one indicates an under proportional share of robotics, greater than one indicates an over proportional share and thus a specialisation in robotics in that spatial unit [56]. To avoid the well-

known small number problem for RTA analyses (see, e.g., [35]) and to ensure a meaningful comparison between urban areas, the data set was reduced to urban areas that show a total patent count greater than 2,000 (see Table A2 in File). Furthermore, spatial units that had a robotic patent count of zero were removed from the data set.

Turning from R&D hotspots and specialisation patterns to R&D collaboration networks, we employ–following recent related works–a Social Network Analysis (SNA) perspective to characterise their structure and dynamics. In general, SNA has come into fairly wide use for the analysis of social systems, originally mostly at the level of socially interacting individuals, but recently also at the level of interacting spatial entities, such as the characterisation of internationalisation trends in networks of R&D collaborations across regions or countries (see, e.g., [14, 27]). This is usually done by aggregating individual level information (in our case inventors) on collaborations to the regional level and shifting attention–away from the traditional variable-centric approach–to a structural-relational angle.

In our analytical approach, we initially define the network under consideration. Graph theory sets out the basic mathematical framework to formally describe our global R&D collaboration network. In our case, we define a graph $G = (N, L, V)$ with $N = \{N_1, N_2, \ldots, N_g\}$ being a set of nodes (here urban areas) which is related through a set of edges $L = \{L_1, L_2, \ldots, L_M\}$ and a set of weights $V = \{V_1, V_2, \ldots, V_M\}$ for each edge, in this study the number of co-patents between two urban areas. The topology of a graph can be decoded in the $n$-by-$n$ adjacency matrix:

$$X_t(i,j) = \begin{pmatrix} x_{11} & x_{12} & \cdots & x_{1n} \\ x_{21} & x_{22} & \cdots & x_{2n} \\ \vdots & \vdots & \ddots & \vdots \\ x_{n1} & x_{n2} & \cdots & x_{nn} \end{pmatrix} \tag{2}$$

where one element of $X$ corresponds to the number of joint co-patents between urban areas $i$ and $j$ at time $t$. The total number of neighbours, i.e., partner countries of a node, is referred to as *degree* of this node. With the adjacency matrix as defined by Eq (2), we can derive a number of relevant global (describing the network structure as a whole) and local (describing the role and positioning of individual nodes) network analytical measures. The following global network measures are used (for a comprehensive and mathematical description, see [14, 59] in a similar context):

- The *average degree* is defined as the sum of the nodes' individual degrees divided by the total number of nodes in the network, indicating that a higher average degree relates to a higher connectedness and integration in the network.

- The *density* is another indicator for the connectedness of a network, defined as the ratio between the number of edges and the highest possible number of edges, ranging between 1 (very densely connected, high integration) and zero.

- The *average path length* is defined as the on average shortest path between all pair of nodes. A path is a sequence of different edges and nodes that connects two nodes with each other, and its length is determined by the edges the path contains. The lower the path length (driven by prominent nodes, so-called hubs), the more cohesive a network is, and the more efficient is the knowledge flow due to shorter ways from one node to another.

- *Clustering* is an indicator that describes the number of "cliques" in a graph, i.e., closed triangles of nodes producing strongly connected sub-graphs and localised knowledge pools.

- *Degree centralisation* describes the centralisation of the entire network or in other words the concentration of the links on certain nodes. There are two extremes: (a) star-like networks with a value of 1 and (b) fully connected graphs with a value of 0. In a star-like network, one actor has the most central position in the network while the other peripheral actors have no or little connections between each other. This central actor is often characterised as a technological gatekeeper who enforces innovative processes and the communication flow significantly. The opposite of a star-like network is a fully connected graph where all actors are connected with each other.

As local measures for identifying the local positioning of individual urban areas, we use three measures pointing to different roles of the different urban areas in the network:

- The *degree-based centrality* is just the normalised degree of a node, a high value indicating an influential position in the network.

- The *betweenness centrality* of a node captures the relative amount of shortest paths going through this node. It is calculated by the sum of the total number of shortest paths between two nodes divided by the sum of the number of those paths that pass through the node. Actors that show a high betweenness are able to contribute global knowledge into the network and have considerable control and power over the information flow. These actors are particularly important in a network as they bridge actors or groups that otherwise would be disconnected.

- The *eigenvector centrality*, also referred to as prestige centrality, is defined as the extent that links a node to other nodes that are central in the network. Actors with high eigenvector centrality are well established, prestigious connections to other influential and central actors.

All SNA indicators–both global and local–have been calculated using the R igraph package (Documentation and download under https://igraph.org/r/). For visualisation purposes, the open-software Gephi was used which helps to display large graphs and reveals patterns and trends in the network. More specifically, the Fruchterman-Reingold Algorithm, which is a force-directed layout algorithm, was used to display the network. Here, the node size corresponds to the weighted degree centrality of a node, while the thickness of the line corresponds to the number of co-patents between them.

## 4 R&D hotspots and specialisation patterns

This section shifts attention to the empirical results of the study, initially reflecting on the illustration and interpretation of the R&D hotspots and specialisation analysis based on the RTA measures as defined in the previous section by Eq (1). Table 1 presents the top ten R&D hotspots and most specialised urban areas in robotics R&D worldwide for the period 2002–2016. The RTA indices (most right column) are put into perspective against the total number of patents for the earliest (2002–2006) and the latest (2012–2016) time period under consideration, with a calculated growth rate between the two periods. Overall, the empirical results remarkably confirm the enormous general growth trend of robotics R&D over the past two decades. In fact, we can observe a quite substantial increase in robotic patenting activity as growth rates mostly show values above 100%, and this growth is also much above the average growth of all patents in an area (across all technologies). In absolute numbers, patenting has increased from 3,262 to 8,531 patents between the first and last period observed in this study. This equals a growth of 162% in absolute terms. Looking at the share of robotic on total patents (to account for the overall increase in patenting), we can find that this share increased from 0.13% in the first reporting period to 0.33% in the last reporting period by 148%. This growth is actually

**Table 1. R&D hotspots in robotics (top ten) and their relative specialisation (RTA).**

| Area* | Number of patents (robotics) | | | Patent growth robotics (in %) | | RTA | | |
|---|---|---|---|---|---|---|---|---|
| | 2002–2006 | 2007–2011 | 2012–2016 | 02/06-07/11 | 07/11-12/16 | 2002–2006 | 2007–2011 | 2012–2016 |
| Ann Arbor | 34.28 | 55.84 | 99.60 | 62.92 | 78.36 | 24.27 | 16.97 | 20.09 |
| Stockholm | 13.80 | 94.12 | 190.76 | 582.00 | 102.69 | 5.43 | 12.98 | 19.05 |
| Detroit | 147.90 | 281.00 | 325.54 | 89.99 | 15.85 | 25.36 | 23.38 | 17.57 |
| Other Aichi | 132.27 | 275.98 | 399.18 | 108.65 | 44.64 | 9.50 | 8.91 | 14.38 |
| Kitakyushu | 7.12 | 36.90 | 88.40 | 418.50 | 139.57 | 1.22 | 4.70 | 13.35 |
| Stuttgart | 232.00 | 379.00 | 384.84 | 63.36 | 1.54 | 19.18 | 15.15 | 13.17 |
| Karlsruhe | 16.72 | 31.44 | 49.35 | 88.03 | 56.99 | 8.66 | 8.40 | 11.32 |
| Goeteborg | 51.92 | 58.02 | 75.86 | 11.76 | 30.74 | 17.09 | 10.17 | 11.28 |
| Munich | 75.76 | 121.27 | 229.88 | 60.07 | 89.55 | 7.69 | 6.90 | 10.40 |
| Frankfurt | 14.96 | 64.35 | 68.31 | 330.19 | 6.14 | 3.98 | 8.62 | 8.83 |

Notes: *Top ten based on RTA in period 2012–2016; area abbreviations are provided in Table A2 in File; calculations include only areas that have a total patent count greater than 2,000; number of patents per area is based on fractional counting of inventors, i.e. the patent is divided equally among all areas in which inventors of the patent are located (e.g. if a patent with two inventors located in two regions A and B, it counts a fraction of 0.5 for both regions).

also remarkable compared to other emerging sectors, e.g., as compared to all ICT (+15%), digital communication technologies (+64%), semiconductors (+7%) or the also very fast-growing technologies related to artificial intelligence (+121%). Note that this is the case for all of the 160 urban areas included in the RTA analysis (see Table A2 in File for an overview on all RTA indices for each urban area included).

Turning to the R&D hotspots, the RTA values indicate that the top ten areas have considerable specialisation advantages. The values range from 8.83 to 20.09, indicating that there are some areas in a country that excessively focus on robotic R&D activities, and that these spatial concentration tendencies have become even more pronounced in the recent past. Only for two areas (Detroit and Stuttgart), the RTA values have slightly decreased between the time period 2007–2011 and 2012–2016, while for 8 areas under the top ten the specialisation advantage has increased even more–most notably Stockholm which had the largest growth in specialisation, more than doubling the RTA between all time periods under consideration. An enormous increase in specialisation is also identified for the Japanese area Kitakyushu that has leveraged its RTA value from 1.2 to 13.3.

Some findings in terms of the top R&D hotspots identified are particularly interesting:

- Ann Arbor (in the US) has the highest RTA index while having one of the lowest robotic patent counts, i.e.it is a highly specialised area in robotics dedicating the majority of its resources in robotic R&D activities. Clearly, that is strongly related to its geographical proximity to Detroit, one of the traditional global robotics hotspots.

- Stockholm (in Sweden) and Kitakyushu (in Japan) show, with growth rates above the thousand percentage mark, by far the strongest development in robotic patenting. In combination with the highest growth observed in the RTA index, these two urban areas clearly shifted their focus to robotic R&D activities and expanded their specialisation advantage considerably over the observed time period.

- Among the top ten, we find regions from the US, Germany, Japan and Sweden. German and US regions have lower growth in comparison to Stockholm and Kitakyushu, being already very active in robotic patenting.

To get a better understanding of the activities and driving forces of the specialisation advantages in the top ten RTA areas, Table A3 in File shows the top three actors (i.e., companies, research institutions or universities) who are the most active and present in robotic patenting between 2012 and 2016 in the top ten areas. In general, companies in the automotive industry are the most dominant patent applicants in robotics, for instance, Ford Global Technologies, GM Global Technology Operations, Daimler, and Toyota. While most urban areas show the highest amounts of patents applied for by automotive companies, Stockholm and Kitakyushu, that have also the highest growth rates in terms of patent activity and RTA, interestingly show different industry orientations. In Stockholm, two of the top three companies (Aktiebolaget Electrolux and Husqvarna) operate in the household, and agriculture and forestry industry. The area Kitakyushu is with Yaskawa Denki–as the most active patent applicant–highly specialised in the production and distribution of industrial robots. Furthermore, it is the only top ten RTA area where a research organisation (National Institute of Advanced Industrial Science and Technology) is among the top three robotic applicants.

The assumption that thematic changes are related to new specialised and emerging areas can be underlined by looking at the evolution of patenting in the robotics CPC subclasses. Indeed, emerging urban areas tend to patent in other CPC robotics classes (mainly B25 classes) compared to the established ones who patent mostly in classical automotive ones (in particular B60). Looking at the importance of CPC subclasses as a whole, this can be supported going beyond the top ten areas. We find that the B25 increase their share in total robotics patenting from 10% (2002–2007) to 15% (2012–2016) between 2002 and 2016.

Considering organisation types, public research organisations are not very active in robotic patenting, as expected from previous works and underlying rationales in R&D processes. However, areas in China and South Korea–especially Beijing, Shanghai, Guangzhou, and Seoul–have a comparatively high share of research facilities operating in robotic patenting, pointing to a more publicly followed strategy to promote robotics R&D. Under the top ten in these four areas, we find for instance the Chinese Academy of Sciences, the Beijing Institute of technology, Jiangsu university or Korea Institute of Science and Technology. This is an interesting pattern and outcome of the empirical analysis, pointing to a more early-stage phase in the evolutionary path of these countries as compared to 'traditional' robotics countries like the US, Germany or Japan, with public R&D being more prevalent.

The R&D hotspot analysis not only reveals the most active and specialised locations in robotics R&D worldwide, but also points to a very dynamic evolution, indicating at least partially a spatial shift in robotics R&D. To underpin this assumption, Fig 1 provides comprehensive information on the developments and movements of specialisation advantages at an area level. It correlates the RTA rank of areas from the second period (2007–2011) to the RTA rank of areas from the third period (2012–2016), depicting movements of ascending (above the line) and descending (below the line) urban areas between the considered periods. Labelled areas are those who have won more than 60 ranks, and those who lost more than 28 ranks. The average rank change is positive (12.08), indicating that there are on average more urban ascending areas than descending ones, i.e., the number of areas specialising in robotics R&D is clearly increasing. This dynamic is highly interesting, because the shift is by far not equally distributed in geographical space. The average RTA increases, while this average increase is subject to new emerging areas, in particular in China, that have not been specialised in the earlier years of the observed time period.

While a specific fraction of regions, in particular at the top of the hierarchy, stayed relatively close to their position in previous periods (underlined by a rank correlation coefficient of 0.82 between the two periods), the most noteworthy observations reflecting interesting dynamics are the following: *First*, there are five Chinese urban areas strongly ascending, showing the

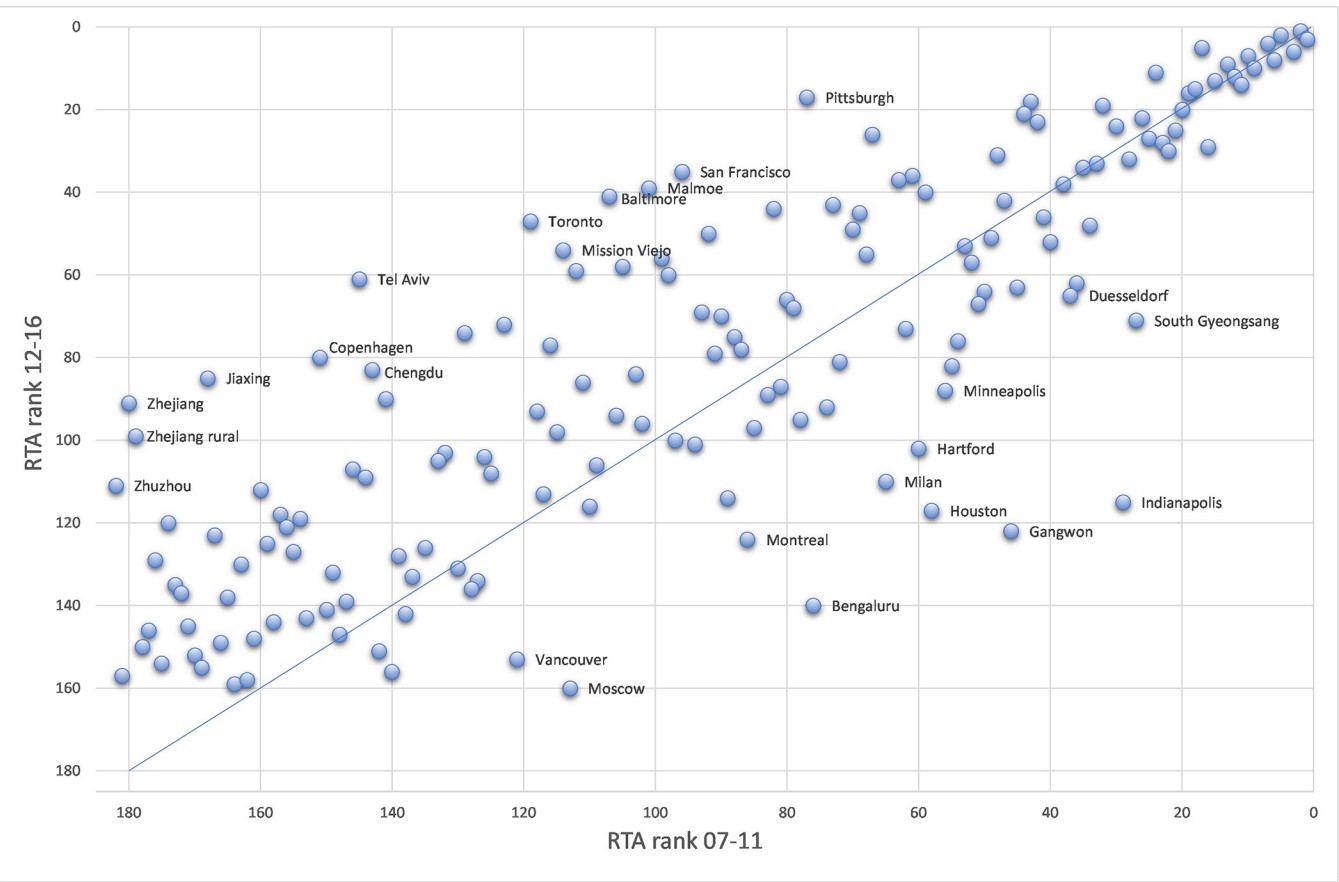

**Fig 1. Dynamics of RTA rankings.**

highest development in robotic specialisation developments. *Second*, there is a shift from national hotspots to other areas in the US. For instance, Indianapolis, Houston, and Minneapolis lost more than thirty ranks, while Pittsburgh, San Francisco, and Baltimore won more than sixty ranks. *Third*, some areas and their movements are particularly striking. Tel Aviv's RTA index increased from 0.056 in 2007–2011 to 0.935 in 2012–2016 (see Table A2 in File), which is an immense growth and shows again how important robotic technology has become in recent years. In contrast, Moscow's RTA index decreased from 0.286 in 2007–2011 to 0.037 in 2012–2016.

## 5 Dynamics of global robotics R&D collaboration networks

R&D activities are–as discussed from a theoretical perspective in Section 2 –nowadays widely accomplished within a web of interacting actors, increasingly located in different regions, or even countries (see, e.g. [27] for the case of global ICT networks). Accordingly, this section puts emphasis on discussing structures and dynamics of the R&D networks in robotics based on the co-patents between urban areas for the time periods under consideration. We initially reflect on global network measures as described in Section 3, and also on the network visualisation that employs a Fruchterman-Rheingold visualisation algorithms, i.e., placing central nodes in the centre of the visualisation, and placing nodes that have many interactions and a same set of partners nearer to each other.

**Table 2.  Global area SNA indicators in the R&D collaboration network.**

|  | **2002–2006** | **2007–2011** | **2012–2016** |
| --- | --- | --- | --- |
| Number of nodes | 933 | 933 | 933 |
| Number of edges | 885 | 1,279 | 1,623 |
| Degree centralisation | 0.06 | 0.07 | 0.07 |
| Average degree | 1.90 | 2.74 | 3.48 |
| Density | 0.0020 | 0.0029 | 0.0037 |
| Average path length | 4.03 | 4.16 | 3.81 |
| Clustering | 0.23 | 0.27 | 0.23 |

Notes: Area abbreviations are provided in the Table A2 in File

node colour represents country of origin; node size corresponds to the degree, thickness of lines to co-patenting intensity between two areas

Before we turn to the urban area level, we want to take a brief look at the global robotics R&D networks at the level of countries, simply be aggregating links observed at the urban area level to the country level. Fig A1 in File provides the country-by-country network for the first and last time period. Most strikingly, the robotics network between countries become much denser, with many countries entering the network in the latest time period or becoming much more central. Next to Germany, being the most central country in the first time period, the US and Japan mainly dominated the network from 2002–2006. Other countries, such as the Netherlands, UK, France, Sweden, but also India, South Korea and China, have been getting a much more central position in the most previous time period.

However, shifting attention to the networks observed at the level of urban areas enables a much more fine-grained picture, in particular in terms of the geographical dynamics. Table 2 presents the respective global SNA indicators, while Fig 2 visualises the networks for the three time periods. Also, at the level of urban areas, results clearly indicate that the number of collaborations has increased significantly over the respective time period. The number of edges almost doubled its value from 885 to 1,623 ties between the first and last period. Moreover, actors are connected with 3.48 other actors on average in the most recent period. Looking at the time span, one can observe a substantial increase in the number of partners as the value of the average connectedness was at 1.90 in the first period.

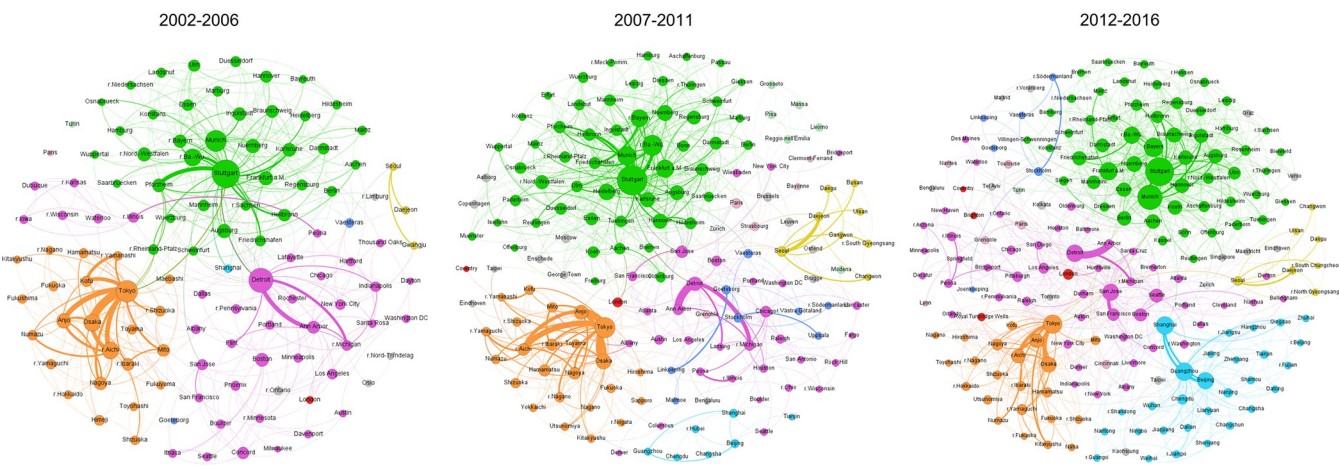

**Fig 2.  Global area R&D collaboration network in robotics.**

The visualisations (Fig 2) together with the top ten urban areas in terms of local SNA indicators (Table 3) help identifying underlying structural mechanisms by looking at specific urban areas and their positioning in the network. They illustrate nicely the changing overall structure of the network on an area level over the observed time periods. The number of nodes (areas) participating in the collaboration network steadily increased from the first period (2002–2006) to the last period (2012–2016). The colour and thickness of the lines suggest that the connections are much stronger within a country than between areas from different countries. For instance, Tokyo has enlarged collaborations with other areas from Japan (e.g., Osaka, Aichi) but only few connections to areas from other countries (and therefore also the relatively decreasing position at the country level, see Fig A1 in File). Because connections are mainly concentrated within a country, colour patterns are visible pointing to some geographical logic

**Table 3. Top ten local area SNA indicators in the R&D collaboration network.**

| 2002–2006 | | 2007–2011 | | 2012–2016 | |
|---|---|---|---|---|---|
| **Area** | **Degree** | **Area** | **Degree** | **Area** | **Degree** |
| Stuttgart | 58 | Stuttgart | 68 | Stuttgart | 67 |
| Tokyo | 54 | Tokyo | 57 | Tokyo | 64 |
| Detroit | 51 | Other Ba.-Wü. | 43 | Munich | 61 |
| Munich | 35 | Munich | 34 | Shanghai | 49 |
| Other Aichi | 33 | Osaka | 33 | Guangzhou | 49 |
| Osaka | 30 | Anjo | 31 | Beijing | 46 |
| Other Ba.-Wü. | 29 | Detroit | 30 | Detroit | 39 |
| Anjo | 27 | Frankfurt a.M. | 30 | San Jose | 38 |
| Boston | 23 | Other Bayern | 28 | Osaka | 38 |
| Other NRW | 21 | Heidelberg | 26 | Other Bayern | 36 |
| **2002–2006** | | **2007–2011** | | **2012–2016** | |
| **Area** | **Between.** | **Area** | **Between.** | **Area** | **Between.** |
| Detroit | 21313.32 | Tokyo | 23231.32 | Detroit | 29933.23 |
| Stuttgart | 13947.73 | Detroit | 15720.42 | San Jose | 19484.21 |
| Tokyo | 12949.14 | Stuttgart | 13770.02 | Shanghai | 19391.76 |
| Paris | 4045.02 | Boston | 10452.12 | Paris | 17987.63 |
| Boston | 3961.02 | Paris | 9506.84 | Tokyo | 17169.19 |
| Lansing | 3770.15 | Anjo | 7897.04 | Munich | 14970.95 |
| Phoenix | 3682.44 | Seoul | 7362.96 | Stuttgart | 13634.90 |
| Santa Barbara | 3496.79 | Other Michigan | 6056.30 | Boston | 10158.62 |
| New York City | 3186.50 | Frankfurt a.M. | 5877.25 | London | 10106.68 |
| Other Illinois | 3070.57 | Chicago | 5866.61 | Guangzhou | 9808.89 |
| **2002–2006** | | **2007–2011** | | **2012–2016** | |
| **Area** | **Eigenvector** | **Area** | **Eigenvector** | **Area** | **Eigenvector** |
| Stuttgart | 1.00 | Stuttgart | 1.00 | Stuttgart | 1.00 |
| Munich | 0.73 | Other Ba.-Wü. | 0.75 | Munich | 0.86 |
| Other Ba.-Wü. | 0.62 | Munich | 0.64 | Other Bayern | 0.70 |
| Tokyo | 0.54 | Frankfurt a.M. | 0.61 | Frankfurt a.M. | 0.64 |
| Detroit | 0.53 | Other Bayern | 0.58 | Other Ba.-Wü. | 0.63 |
| Nuernberg | 0.46 | Nuernberg | 0.53 | Nuernberg | 0.53 |
| Pforzheim | 0.44 | Hannover | 0.49 | Karlsruhe | 0.52 |
| Other NRW | 0.44 | Augsburg | 0.48 | Hannover | 0.52 |
| Other Bayern | 0.43 | Heidelberg | 0.46 | Ingolstadt | 0.51 |
| Frankfurt a.M. | 0.41 | Karlsruhe | 0.46 | Augsburg | 0.50 |

in the network. Areas from Germany, China and Japan are grouped together as areas from these countries are mainly connected to one or few central hubs within the country that are further connected to international areas; a pattern that can be related to the global pipelines versus local buzz considerations in the literature (see e.g. [60]).

In the US, there are more areas that show international connections which is why areas from the US are spread through the network structure. Moreover, urban areas from Sweden are moving closer to each other, and also closer to urban areas from Germany. This indicates that the collaboration between areas from Sweden and Germany intensified while collaborations between areas from Sweden and the US weakened, at least in relation. The rise of Chinese urban areas is striking when comparing the visualisations of all three time periods. While Shanghai was the only urban area present in the first period, there are more than twenty areas from China illustrated in the network in the most recent time period. This is a significant and noteworthy change in the network structure as Chinese cities are now one of the dominant actors and no other country can show this intense development.

Shifting attention to the top urban areas in terms of network centrality (see Table 3), we follow previous research on R&D collaboration networks (see, e.g. [6, 14, 61]) and look–as described in Section 3 –at the local interaction intensity of a node (urban area) in the network by means of degree centrality, the global embedding in terms of the connection of a node to other central nodes (eigenvector centrality), and the ability to 'bride' interaction between other nodes (betweenness centrality). While these measures are positively correlated, we still find some interesting differences. For instance, more established areas (e.g., in Germany) tend to have a higher eigenvector centrality (i.e., they are more connected to a long-established core than emerging areas) but are also able to act as 'bridges' in the network (reflected by the high betweenness centrality for US and Japanese areas). In contrast, emerging urban areas (and here in particular Beijing, Guangzhou and Shanghai) appear at the top in more local embeddings (degree centrality), while still somewhat behind the top players in the other quality dimensions.

Turning to individual areas, it can be seen that Stuttgart and Tokyo are the two cities which show constantly the highest amount of connections, indicating that they hold a very influential position in the network. Urban areas which were able to increase their direct connections, such as Munich, Osaka, and Bavaria (Bayern region), improved their integration into the network. In contrast, Detroit and Aichi lost their influential position slightly over time and are overruled by other areas. In fact, three areas from China–Shanghai, Guangzhou, and Beijing–show significant development in terms of direct connections. While their number of direct connections were 6 (Shanghai), 3 (Guangzhou) and 0 (Beijing) in the first period, they all have more than 45 direct connections with other actors in the last period and are hence much better integrated than other European urban areas, for instance. Fig 2 illustrates the reason for this increase: The significant surge of direct connections can be mainly attributed to a rise of other urban areas located in China and national connections to them.

Most changes and movements can interestingly be observed for the betweenness centrality of the top ten actors. There have been major downward and upward developments of actors from the US in terms of global knowledge contribution. In the first period, Phoenix, Santa Barbara, and Illinois region were able to influence and control the knowledge flow in the network significantly. However, all three areas show a downward trend and Santa Barbara even disappears completely from the collaboration network in the second and third period. Instead, the knowledge contribution role was shifted to the areas Boston, Detroit, and San Jose. While Detroit and Boston have been constantly under the top ten betweenness indicators, San Jose suddenly appears as the second influential knowledge contributor in the last period. Moving from the US to Germany, one can observe that Stuttgart has become slightly less responsible

for the global knowledge flow as the betweenness indicator decreases constantly over the time periods, whereas Munich rises as an important actor in the collaboration network. Similar developments can be detected when taking a Japanese perspective: There is less global knowledge flow through Tokyo and more global knowledge flow through other Japanese areas such as Osaka, Anjo or Aichi region. Moreover, the rise of Chinese areas (i.e., Shanghai and Guangzhou) can also be seen in the developments of their betweenness centralities.

Interestingly, German regions are extremely prominent in terms of eigenvector centrality. In fact, all areas listed under the top ten–except from Tokyo and Detroit in the first period–are located in Germany. Furthermore, an extended list of the top fifty areas with the highest eigenvector centrality shows that the majority of areas are located in Germany. Accordingly, German actors in robotics R&D collaboration networks are at a much higher degree connected to other central countries, rather than to more peripheral ones. One the one hand, this shows the strong tradition of Germany in the robotics landscape, but, on the other hand, Germany is less connected with emerging new players that are not central in the network yet but most likely bring in new relevant knowledge elements as well. Hence, missing links to these emerging areas could be a threat for the future R&D capacity of established areas like German ones.

In sum, the analysis shows the quite dynamic character of the global robotics R&D collaboration, with changing roles of several urban areas worldwide, characterised both by ascending and descending ones in terms of network centrality, both within as well as between countries. The overall trend of rising R&D activities and collaborations is striking, as is the importance of spatial shifts driven by technological evolution, but also other intervening factors, e.g., strong public R&D funding for the sector in specific places. While the systematic identification of drivers and more detailed explanations of these observed patterns is out of scope of this study, it provides the basis for future research in this direction. Moreover, it suggests interesting case studies to explain observed intra-national shifts (e.g., within the US) on ascending and declining areas in terms of their network position.

## 6 Discussion and concluding remarks

In recent years, the robotics market has been growing substantially–mainly in manufacturing industries such as the automotives, electronics or pharmaceuticals sectors, but also more increasingly in service industries–and it is expected to grow further in the future. Given its high research intensity in combination with the high growth and transformative potential for the economy and the society, robotics is widely considered as breakthrough innovation, both in the scientific realm as well as by policy makers around the world. As a highly research- and capital-intensive segment, robotics requires a high level of expertise and knowledge as well as sufficient resources which can usually not be covered by single organisations, or even regions or countries. Therefore, R&D collaborations may play an even more important role in the robotics segment, based on insights from innovation studies telling us that the successful generation of new knowledge as a basis for innovation is increasingly conducted within a complex web of interacting researching actors (firms, universities, research organisations, intermediaries, agencies, etc.). Moreover, given the high public investments in robotics R&D, the global landscape is assumed to change dynamically, both in terms of its geographical distribution and thematic orientation. However, systematic empirical insights into the geographical dynamics of global robotics R&D are scarce so far.

Against this background, the aim of this study was to identify and characterise the changing R&D landscape in robotics, for the first time globally and through very detailed geographical lenses of global urban areas. We trace global R&D hotspots and characterise geographical specialisation patterns in robotics worldwide, as well as dynamics of underlying R&D

collaboration networks. Patents are used as marker for commercially oriented, technological R&D activities, and co-patents for collaborative inventive activities, respectively. Our sample dataset comprises more than 41,000 inventor locations of patents applied for from 2002–2016, shifting attention to three time periods for tracing dynamics (2002–2006, 2007–2011 and 2012–2016). We identify R&D hotspots and the geography of specialisation in robotics through the Revealed Technological Advantage (RTA) index, while we employ a Social Network Analysis (SNA) perspective to characterise robotics R&D collaboration networks based on the identified co-patents.

Condensing the empirical results, we discovered some substantial new insights about the global geography of the robotics R&D landscape: *First*, we identify most prominent urban hotspots in terms of specialising in robotics R&D; the most specialised urban areas are located in Germany, the US, Japan, but also Sweden. However, geographical specialisation patterns change, and this shift is by far not equally distributed in geographical space; the increase in specialisation advantages is to a substantial extent subject to new emerging areas, in particular in China. *Second*, we find that global robotics R&D has grown immensely–compared to overall global patenting but also to other high growth technologies–both in total patenting and in terms of R&D collaboration activities between urban areas. The global networks density and average number of collaboration partners has nearly duplicated between the periods 2002–2006 and 2012–2016. *Third*, the growth related to the network is spatially by far not equally distributed, with some cities remarkably increasing their central positioning, while others decline in terms of network centrality. In the latter context, most strikingly a high number of Chinese urban areas are entering the network, with three cities (Beijing, Shanghai and Guangzhou) now belonging to the globally most central urban areas in robotics R&D. Interestingly, this resembles a local buzz vs global pipelines structure of global R&D collaboration networks (see, [60]), with these three areas acting as important national knowledge 'gatekeepers', distributing knowledge into the Chinese system. Next to these Chinese areas, the network is mainly dominated by urban areas located in Germany (e.g., Stuttgart, Munich), the US (e.g., Detroit, San Jose) and Japan (e.g., Tokyo, Osaka). *Fourth*, the observed spatial shifts can be to a substantial degree attributed to changing thematic priorities in robotics R&D. While we find the classical automotive oriented cities exclusively on top in the earliest period, other fields of application (mainly under the B25 class), such as household, agriculture and forestry, are at stake under the most growing areas in terms of specialisation but also network centrality (Stockholm and Aichi). Path-dependence of established hotspots may be at stake here enabling other cities to step in by diversifying into new innovative sub-branches, being one potential driver of the observed changing network structures. *Fifth*, we interestingly find a stronger role of public research in Asia than in other territories which are active in robotics R&D. This could be related to the fact that public R&D funding of robotics played a substantial role in those countries, in particular China. *Sixth*, intra-national shifts are remarkable and visible by the analysis; for instance, in the US, some urban areas have strongly increased their network centrality (e.g., San Jose), while others lose their position (e.g., Santa Barbara).

In relation to theoretical debates in economic geography, regional science and network science, certain assumptions from a theoretical perspective are underlined, bringing an interesting empirical perspective to them. *First*, the study empirically confirms evolutionary pathways expected from a theoretical perspective, that is a growing internationalisation and globalisation of R&D–widely discussed in economic geography literature (see, e.g., [62])–but at the same time geographical localisation remaining prevalent (see, e.g., [4]). Although the robotics network is highly dynamic, spatial proximity stays relevant, given that the majority of links still occur between cities that are geographically close to each other, and this is due to the necessity for exchanging tacit knowledge components–especially in knowledge intensive segments like

robotics–that are costly to be transferred in geographical space. *Second*, we can observe specific network structural mechanisms discussed in related literature inspired from a social network perspective (see, e.g., [7]), such as the growing integration and increasing connectedness of social systems over time, but also the structuring of such networks by changing roles of nodes in terms of their network positioning. *Third*, the study provides conceptual inputs to the debate on regional diversification and path-dependence [63, 64], showing that some cities are able to branch into new technologies and diversify their productive structure. This may be one important driver of the underlying changing network structure and positioning of specific cities observed in this study.

From a policy perspective, it is worth noting that public R&D funding instruments presumably played a strong role for the catching-up process of Asian countries, in particular China. What is somewhat surprising is that this catching-up is even more pronounced in the changing network structure, where Asian cities tend to get in very central position, and therefore, a broader view of the knowledge is being circulated. Accordingly, shifting attention only to the overall amount of R&D support in policy practice at country or supra-national level falls far too short, given that regional diversification processes into related technologies play an important role, and can be assumed to be one driver of the changing network structures observed in this study.

Considering limitations of this study, some ideas for future research come into mind. *First*, the study provides a comprehensive empirical view on global dynamics in robotics R&D and points to interesting drivers (e.g., public R&D funding, spatial and other forms of proximity, regional diversification and branching, network structural mechanisms). This opens the perspective to statistically investigate the magnitude and significance of these drivers and their interplay, for instance in a spatial interaction modelling or inferential network modelling approach (e.g., temporal exponential random graph models). *Second*, changing thematic orientations and application fields, for instance by looking at specialisations and networks of specific subdomains of robotics, are of great interest for follow-up research. In relation to this, the geography of the basic research activities–underlying the technological developments in robotics as captured by patents–would be interesting to be investigated, for instance by looking at scientific publications or R&D projects and comparing their space-time patterns to patenting activities.

## Supporting information

**S1 File. Data extraction and preparation.**
(DOCX)

## Author Contributions

**Conceptualization:** Thomas Scherngell.

**Data curation:** Thomas Scherngell, Katharina Schwegmann, Georg Zahradnik.

**Formal analysis:** Thomas Scherngell, Katharina Schwegmann, Georg Zahradnik.

**Funding acquisition:** Thomas Scherngell.

**Investigation:** Thomas Scherngell.

**Methodology:** Thomas Scherngell, Katharina Schwegmann, Georg Zahradnik.

**Project administration:** Thomas Scherngell.

**Supervision:** Thomas Scherngell.

**Validation:** Thomas Scherngell, Katharina Schwegmann, Georg Zahradnik.

**Visualization:** Thomas Scherngell, Katharina Schwegmann.

**Writing – original draft:** Thomas Scherngell, Katharina Schwegmann.

**Writing – review & editing:** Thomas Scherngell, Katharina Schwegmann, Georg Zahradnik.

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
