## [Decision Letter · Decision Letter 0]

10 Mar 2022

PONE-D-21-39533The geographical dynamics of global R&D collaboration networks in robotics: Evidence from co-patenting activities across urban areas worldwidePLOS ONE

Dear Dr. Scherngell,

Thank you for submitting your manuscript to PLOS ONE. After careful consideration, we feel that it has merit but does not fully meet PLOS ONE’s publication criteria as it currently stands. Therefore, we invite you to submit a revised version of the manuscript that addresses the points raised during the review process.

As point out by the two reviewers, the paper is weak on numerous points specifically: how do you define and delineate your cities (many nodes seem to not be cities even)? You should justify stronger the focus on Robotics, situate better in the large literature on evolutionnary economic geography treating these patents' collaborations. I acknowledge the huge work made by the two reviewers : see all the details below. Your paper really needs a large conceptual refundation, numerous corrections and deep reformulations of the interpretations.

We look forward to receiving your revised manuscript.

Kind regards,

Celine Rozenblat

Academic Editor

PLOS ONE

Journal Requirements:

“This was supported by RISIS2 (Research Infrastructure for and Innovation Policy Studies 2), funded by the European Union’s Horizon2020 Research and innovation programme under the grant number n°824091”.”

“TS, KS, GZ: partly funded by the European Union under Horizon2020 Research and Innovation Programme Grant Agreement n°82409; https://ec.europa.eu/info/funding-tenders/opportunities/portal/screen/programmes/h2020; funder played no role in study design”

“TS, KS, GZ: partly funded by the European Union under Horizon2020 Research and Innovation Programme Grant Agreement n°82409; https://ec.europa.eu/info/funding-tenders/opportunities/portal/screen/programmes/h2020; funder played no role in study design”           

“NO authors have competing interests”

7. Please upload a copy of Figures 4 and 5 to which you refer in your text on page 15. If the figure is no longer to be included as part of the submission please remove all reference to it within the text.

Additional Editor Comments (if provided):

As point out by the two reviewers, the paper is weak on numerous points specifically: how do you define and delineate your cities (many nodes seem to not be cities even)? You should justify stronger the focus on Robotics, situate better in the large literature on evolutionnary economic geography treating these patents' collaborations. I acknowledge the huge work made by the two reviewers : see all the details below. Your paper really needs a large conceptual refundation, numerous corrections and deep reformulations of the interpretations.

Reviewers' comments:

Reviewer's Responses to Questions

**Comments to the Author**

1. Is the manuscript technically sound, and do the data support the conclusions?

Reviewer #1: Partly

Reviewer #2: Partly

2. Has the statistical analysis been performed appropriately and rigorously? 

Reviewer #1: Yes

Reviewer #2: Yes

3. Have the authors made all data underlying the findings in their manuscript fully available?

Reviewer #1: Yes

Reviewer #2: No

4. Is the manuscript presented in an intelligible fashion and written in standard English?

Reviewer #1: Yes

Reviewer #2: Yes

5. Review Comments to the Author

Reviewer #1: The paper ‘The geographical dynamics of global R&D collaboration networks in robotics: Evidence from co-patenting activities across urban areas worldwide’ focuses on the R&D collaboration in robotics, which is innovative in the research scale and industry. However, there are several questions needed to be answered:

1. What is the reference to choose the particular CPCs of robotics?

2. What is the reference of choose CPCs of robotics? Although robotics is a knowledge intensive industry, the basic technologies also matter. What is the role of the fundamental technologies?

3. How to define the concept of urban area? What is the delineation? And What is the standard to choose the 900 urban areas?

4. In note 4 (p.7), it is pointed out that ‘rural areas are included in the data, but patents are almost exclusively recorded to urban areas.’. However, in Tab.1, Aichi Rural is on the forth position. Does it reflect the inconsistency of the data sample?

Reviewer #2: Thank you for giving me the opportunity to read this study, in which you propose an interesting empirical case on the geographical localization of robotic patents. I think that your paper makes a timely and relevant contribution to make sense of the spatial emergence of this new industry that is already having profound impacts on the economy. However, I noticed a number of drawbacks to address, of which I believe the most urgent to tackle is to better ground your contribution onto the existing literature on the geography of innovation. Good luck with the revision!

1- Introduction

1.1-The growth of the robotics industry is presented as an opportunity for economic growth and competitiveness. However, there is no acknowledgement of the discussion about the societal challenges implied by it, particularly in terms of job losses (Frey and Osborne, 2017; Acemoglu and Restrepo, 2020), but more widely in the whole re-organization of societal functions and their impact in urban life (Macrorie et al., 2019). I understand that these discussions are not your key concern; however, you say that robots could address important societal challenges, but at the same time you might acknowledge that they could generate other problems.

1.2- The objectives of the study are somewhat general and not very connected to a gap in the literature. You say at page 3 that: “few studies exist that systematically trace the global landscape of robotic innovations and the related global R&D collaboration networks behind it”. Yet this does not explain why we need more studies on this, and what you would add to the existing literature. The fact that robotics are a very innovative technology is not sufficient, as other technologies could also be interesting to study including bio-engineering, artificial intelligence, internet of things or others. Also, many studies have geocoded patents, so I think you should play down the extent to which this is new, or explain how it differs from other contributions in this respect (see for example: Morrison et al., 2017; de Rassenfosse et al., 2019; Miguelez et al., 2019).

1.3- Audretsch and Feldmann, 1996: Reference is missing in bibliography

2- Literature review

2.1- Related to the previous remark, the paper should be better embedded in the relevant literature. In fact, you discuss mainly applied contributions to the topic of robotics, but your motivation for the study is to understand the geographical dimension of R&D activities in this sector. You mention collaboration networks and knowledge exchange dynamics, and there is a huge literature on economic geography and the geography of innovation of which you could at least mention some of the main traits, authors and debates. I can’t resume here this literature but it includes discussions on the extent to which different forms of proximity favor networking, on knowledge complexity, on regional branching and relatedness and many other topics (you might check Balland and Boschma, 2021 for a recent application of some of these concepts on Industry 4.0 technologies, including robotics). These approaches are likely not all necessary for your point, but I believe you should ground your contribution onto some conceptualization of R&D and innovation dynamics in space, that is now lacking, and the related debates to which your paper contributes.

3-Methods

3.1- Perhaps you could explain better the definition of urban areas that is used, or refer to an established definition (OECD?).

3.2- In the RTA formula, the right part at the denominator you sum over i twice, whereas you should sum over i and k. Also, it might be clearer if you put it graphically this way:

(From Boschma et al., 2015).

3.3- Page 8 last paragraph, the following phrase has some syntactic problem (missing verbs?):

In general, SNA has come into fairly wide use for the analysis of social systems, been to be interpreted at the individual level of socially interacting individuals

3.4- At page 9, first paragraph, adjacency matrix as defined by equation (1 2)

3.5- At pages 9 and 10, the lists of network measures is a bit too long and probably not necessary as they are fairly well known. You could refer to Wasserman and Faust for a full definition and just list the measures used and what they show in general, using half the space.

4- Results

4.1- Table 1 is unclear. In fact, while the title is “RTA analysis”, the first three fields are not RTA indexes, because RTA appears only in the last field. This raises two questions: if the first three fields represent patent counts, why are they not integer numbers? If you have used fractional counts, assigning a proportional contribution to inventors in different cities, you should mention it before, so the reader can interpret these results. Second question: how can you say anything about the evolution of specialization if you only show RTA for the period 2012-2016 and not the previous one? The increase in absolute patent numbers could be simply an indication of the aggregate growth of patenting activity.

4.2- Related to this, I have some doubts that these results can support the findings you report at page 11. Particularly, in finding number 2 you say that two areas have shifted their focus to robotic R&D, but even though these numbers could suggest it, they are not enough to state it confidently. These data only say part of the story, so that for example, this might be part of a common growth in robotic specialization, or other technology fields in these urban areas might be growing even more than robotics. Hence, I suggest to nuance a bit more these statements.

4.3- Page 12, first paragraph: you say that “most urban areas show a strong specialization in the automotive industry”, but what you want to say is that applicants in these regions are mostly automotive companies. Your data do not allow you to say much about the overall specialization of these areas.

4.4-Same page-same paragraph as above, why would you expect universities or research organization to be at the top of the list? Also, data about higher share of research facilities in the Asian cities you mention is not disclosed.

4.5- The table A5 you mention at page 11 does not exist in the appendix. Is it A3?

4.6- Figure 1 and the related commentaries suggest that there has been a general increase more than decrease of specialization in robotics (related to commentary 4.2). This should be acknowledged somewhere.

5-Network results

5.1- In figure 2, in the notes it is not clear to what “number” link size refers to.

5.2- There doesn’t appear to be any table A8 or figure 5.

5.3- Page 17, third paragraph: “places on earth” sounds vague.

5.4- In this section you use three different measures of centrality. To which in particular do you refer when you talk of “network centrality” (p.15)? Do you consider them as being equally important or you give priority to one over others? What is the value-added of considering different measures of centrality?

6- Discussion

6.1- Page 18 last paragraph, you say that robotics “has grown immense”. However, you should contextualize this finding, because we know that patenting in general has greatly increased in past decades. At the same time, growth in robotics could be relativized against growth in other emergent sectors. Otherwise, this claim does not appear very substantiated.

6.2 –I do not see how the limited evidence presented in table A3 can support such vast claim as the fourth one. The remark is interesting, but you should support it with further evidence, perhaps including a larger number of patent applicants and extending research beyond the top 10 RTA areas.

6.3- Data in support of the fifth claim is not presented (see comment 4.4).

6.4- The discussion does not explain how research findings relate to existing debates and research directions in the literature (see also comments 1.2 and 2.1).

References

Acemoglu D, Restrepo P. Robots and Jobs: Evidence from US Labor Markets. journal of political economy. : 57.

Balland P-A, Boschma R. Mapping the potentials of regions in Europe to contribute to new knowledge production in Industry 4.0 technologies. Regional Studies. 2021;55: 1652–1666. doi:10.1080/00343404.2021.1900557

Boschma R, Balland P-A, Kogler DF. Relatedness and technological change in cities: the rise and fall of technological knowledge in US metropolitan areas from 1981 to 2010. Industrial and Corporate Change. 2015;24: 223–250. doi:10.1093/icc/dtu012

de Rassenfosse G, Kozak J, Seliger F. Geocoding of worldwide patent data. Sci Data. 2019;6: 260. doi:10.1038/s41597-019-0264-6

Frey CB, Osborne MA. The future of employment: How susceptible are jobs to computerisation? Technological Forecasting and Social Change. 2017;114: 254–280. doi:10.1016/j.techfore.2016.08.019

Macrorie R, Marvin S, While A. Robotics and automation in the city: a research agenda. Urban Geography. 2021;42: 197–217. doi:10.1080/02723638.2019.1698868

Miguelez E, Raffo J, Chacua C, Coda-Zabetta M, Yin D, Lissoni F, et al. Tied in: the Global Network of Local Innovation. : 55.

Morrison G, Riccaboni M, Pammolli F. Disambiguation of patent inventors and assignees using high-resolution geolocation data. Sci Data. 2017;4: 170064. doi:10.1038/sdata.2017.64

6. PLOS authors have the option to publish the peer review history of their article (what does this mean?). If published, this will include your full peer review and any attached files.

Reviewer #1: No

Reviewer #2: No

---

## [Decision Letter · Decision Letter 1]

4 Sep 2022

PONE-D-21-39533R1The geographical dynamics of global R&D collaboration networks in robotics: Evidence from co-patenting activities across urban areas worldwidePLOS ONE

Dear Dr. Scherngell,

Thank you for submitting your manuscript to PLOS ONE. After careful consideration, we feel that it has merit but does not fully meet PLOS ONE’s publication criteria as it currently stands. Therefore, we invite you to submit a revised version of the manuscript that addresses the points raised during the review process.

Your article was significantly improved. However, it still misses some important links to the literature. Reviewer 2 gives you some very kind insights to reach this last step. I hope you will appreciate.

We look forward to receiving your revised manuscript.

Kind regards,

Celine Rozenblat

Academic Editor

PLOS ONE

Journal Requirements:

Additional Editor Comments:

Your article was significantly improved. However, it still misses some important links to the literature. Reviewer 2 gives you some very kind insights to reach this last step. I hope you will appreciate.

Reviewers' comments:

Reviewer's Responses to Questions

**Comments to the Author**

1. If the authors have adequately addressed your comments raised in a previous round of review and you feel that this manuscript is now acceptable for publication, you may indicate that here to bypass the “Comments to the Author” section, enter your conflict of interest statement in the “Confidential to Editor” section, and submit your "Accept" recommendation.

Reviewer #1: All comments have been addressed

Reviewer #2: (No Response)

2. Is the manuscript technically sound, and do the data support the conclusions?

Reviewer #1: Yes

Reviewer #2: Yes

3. Has the statistical analysis been performed appropriately and rigorously? 

Reviewer #1: Yes

Reviewer #2: Yes

4. Have the authors made all data underlying the findings in their manuscript fully available?

Reviewer #1: Yes

Reviewer #2: Yes

5. Is the manuscript presented in an intelligible fashion and written in standard English?

Reviewer #1: Yes

Reviewer #2: Yes

6. Review Comments to the Author

Reviewer #1: The detailed response from the authors appropriately correspond to the previous questions and comments. The manuscript is modified, the appendix is updated and the reference is adjusted according to the comments from the two reviewers. This study basically meets the criteria of publication.

Reviewer #2: Thank you for submitting this revised version of your paper. Your modifications have improved the paper substantially, and mostly address my previous remarks. However, I still think you need to make an extra effort to anchor your contribution better onto existing literature. Your empirical results are very valuable, and I believe you can connect them to existing theoretical debates, but the link needs to be made more explicit (see remarks 1.2, 2.1, 6.3). Apart from these remarks, the rest are secondary corrections.

1. Introduction

1.1- When you discuss mission-oriented research policies you may want to mention that robotics is part of the Key Enabling Technologies identified by the EU (KETs), and the related definition of General Purpose Technologies (GPT). See Montresor and Quatraro (2017).

1.2- Research objectives are still not connected to a literature debate. I see that you motivated better the fact that robotics is an important industry that has impact on the economy, which makes your paper timely and societally relevant. Yet I still feel as your main goal is describing the robotics industry per se, and not as a case study to investigate geographical specialization or network dynamics. You can have purely empirical objectives, but I am convinced that the paper would benefit if you could make explicit also a theoretical one.

2. Literature review

2.1- Thank you for integrating literature on the societal impact of robotization, and for adding references to debates in economic geography and network science. However, the reader still does not understand how you integrate a geographical and “a-spatial” perspectives, and what you add to existing debates on these topics (e.g. Glückler, 2007; ter Wal and Boschma, 2011). You don’t seem to use the literature you presented to support a specific point beyond the fact that robotics is a strategic industry whose spatial configuration has not been studied.

2.2- This section is rather long, so you might benefit from creating thematic sub-sections of this chapter, to make the argument clearer.

3. Data and methods

3.1- Thank you for clarifying the spatial delimitation and the issue of fractional counts. It’s much clearer now.

3.2- Just before the RTA equation, you write: “Regional specialization in robotics exists when a spatial uses”. You probably forgot something.

4. Results

4.1- Tables are much clearer with the contextualized figures.

4.2- In figure 1, the line looks as a regression line, but not one that identifies, as you say, “movements of ascending (above the line) and descending (below the line) urban areas” in terms of RTA. If you wanted to show that, the line should be the diagonal, with slope coefficient of 1 and intercept 0. You can easily check this by manually considering the points in your figure. For example, the one I highlighted here below should be on a descending trend according to you: instead, its rank is around 150 in period 07-11 and around 130 in 12-16, so rank is ascending. The line I plotted, by contrast, rightly identifies the point as being ascending. Actually, this shows that an even higher share of cities is increasing their specialization in robotics, with respect to your previous line. It also shows that the top cities perfectly align with the line, so that the hierarchy at the top is rather stable.

5. Network dynamics

5.1- The discussion of different centrality measures is very pertinent and insightful, thank you.

5.2- Thank you for clarifying the legend of network images. However, it’s not clear to which interactions correspond the thickness of lines. You cite 5 pages earlier that it’s co-patenting intensity, but a reminder about that also in the graph’s legend would make it clearer.

5.3- Paragraph 4, the last sentence: is it an argument of local buzz vs global pipelines or rather one of gatekeeping? I think it’s more about cities being national gatekeepers with respect to global flows.

5.4- Paragraph 6, the ability to “bridge” instead of “bride”.

6. Discussion and conclusions

6.1- In this section, I suggest you separate more clearly the empirical results of your paper from the conceptual and policy contributions. Right now, you mix empirical results (paragraphs 3 and 5) with conceptual ones (paragraph 4) and policy (paragraph 6). I think you should organize it better by providing empirical results first, then conceptual/policy results.

6.2- Furthermore, I think you should add a brief discussion of the limitations of your study and link it more coherently to a discussion on research perspectives, that right now is a bit lost into policy conclusions.

6.3- In relation to your contribution to the literature, paragraph 4 does not answer to the question: what is the new contribution of this study to the relevant literature? You address some points, but the discussion is still too vague. For example: “a certain logic in terms of proximity dimension”. Which logic do you refer to? Also, the “specific network structural mechanisms” do not look so specific but rather general (growing integration/changing roles of nodes). I understand that your study is descriptive, and that your results partly confirm existing studies, which is fine. However, you could point to the conceptual advances that you provide besides the empirical novelty of the case study. I think that findings on changes in specialization and different forms of centrality are interesting to the debate on regional diversification and path-dependence (Grillitsch et al., 2018; MacKinnon et al., 2019). This relates to the capability of innovative cities to retain their edge or lose it because others are better at acquiring new technological innovations. In fact, you show that German cities are very central in terms of Eigenvector centrality but, for example Stuttgart, loses betweenness centrality. This is an interesting empirical finding that you could link to the debates that are at the core of contemporary economic geography such as the capability of cities and regions to branch into new technologies, (smartly) diversify and renew their productive structure. You could also expand a bit more on insight number four (automotive vs other robotics sector) which could also be a relevant point to path-dependence debates.

References

Gluckler, J., 2007. Economic geography and the evolution of networks. Journal of Economic Geography 7, 619–634. https://doi.org/10.1093/jeg/lbm023

Grillitsch, M., Asheim, B., Trippl, M., 2018. Unrelated knowledge combinations: the unexplored potential for regional industrial path development. Cambridge Journal of Regions, Economy and Society 11, 257–274. https://doi.org/10.1093/cjres/rsy012

MacKinnon, D., Dawley, S., Pike, A., Cumbers, A., 2019. Rethinking Path Creation: A Geographical Political Economy Approach. Economic Geography 95, 113–135. https://doi.org/10.1080/00130095.2018.1498294

Montresor, S., Quatraro, F., 2017. Regional Branching and Key Enabling Technologies: Evidence from European Patent Data. Economic Geography 93, 367–396. https://doi.org/10.1080/00130095.2017.1326810

Ter Wal, A.L.J., Boschma, R., 2011. Co-evolution of Firms, Industries and Networks in Space. Regional Studies 45, 919–933. https://doi.org/10.1080/00343400802662658

7. PLOS authors have the option to publish the peer review history of their article (what does this mean?). If published, this will include your full peer review and any attached files.

Reviewer #1: No

Reviewer #2: No

---

## [Decision Letter · Decision Letter 2]

23 Jan 2023

The geographical dynamics of global R&D collaboration networks in robotics: Evidence from co-patenting activities across urban areas worldwide

PONE-D-21-39533R2

Dear Dr. Scherngell,

We’re pleased to inform you that your manuscript has been judged scientifically suitable for publication and will be formally accepted for publication once it meets all outstanding technical requirements.

Kind regards,

Celine Rozenblat

Academic Editor

PLOS ONE

Additional Editor Comments (optional):

Thanks for the changes according to our suggestions that are very appreciated

Reviewers' comments:

Reviewer's Responses to Questions

**Comments to the Author**

1. If the authors have adequately addressed your comments raised in a previous round of review and you feel that this manuscript is now acceptable for publication, you may indicate that here to bypass the “Comments to the Author” section, enter your conflict of interest statement in the “Confidential to Editor” section, and submit your "Accept" recommendation.

Reviewer #1: All comments have been addressed

Reviewer #2: All comments have been addressed

2. Is the manuscript technically sound, and do the data support the conclusions?

Reviewer #1: Yes

Reviewer #2: Yes

3. Has the statistical analysis been performed appropriately and rigorously? 

Reviewer #1: Yes

Reviewer #2: Yes

4. Have the authors made all data underlying the findings in their manuscript fully available?

Reviewer #1: Yes

Reviewer #2: Yes

5. Is the manuscript presented in an intelligible fashion and written in standard English?

Reviewer #1: Yes

Reviewer #2: Yes

6. Review Comments to the Author

Reviewer #1: The detailed response from the authors appropriately correspond to the previous questions and comments. The modified manuscript makes the study more solid and enriched.

Reviewer #2: Thank you for your corrections and comments. I still feel that the paper is quite dense, but you made a considerable effort to make your argument more explicit and better embedded in the literature, so congratulations.

7. PLOS authors have the option to publish the peer review history of their article (what does this mean?). If published, this will include your full peer review and any attached files.

Reviewer #1: No

Reviewer #2: No

---

## [Editor Report · Acceptance letter]

6 Apr 2023

PONE-D-21-39533R2 

The geographical dynamics of global R&D collaboration networks in robotics: Evidence from co-patenting activities across urban areas worldwide 

Dear Dr. Scherngell:

I'm pleased to inform you that your manuscript has been deemed suitable for publication in PLOS ONE. Congratulations! Your manuscript is now with our production department. 

Kind regards, 

on behalf of

Prof. Celine Rozenblat 

Academic Editor

PLOS ONE